# Assessment of Airflow Patterns Induced by a Retractable Baffle to Mitigate Heat Stress in a Large-Scale Mechanically Ventilated Barn

Seunghyeon Jung, Hanwook Chung  and Christopher Y. Choi *

Department of Biological Systems Engineering, University of Wisconsin—Madison, 460 Henry Mall, Madison, WI 53706, USA; jung65@wisc.edu (S.J.); hchung45@wisc.edu (H.C.)
* Correspondence: cchoi22@wisc.edu

**Abstract:** In large-scale dairy farming, heat stress remains a primary concern, and cross-ventilated barns have become increasingly prevalent in order to tackle this issue. Such barns employ energy-intensive electrical fans to enhance airflow and regulate temperature. To optimize this system, air baffles are often placed above the animal-occupied zones (AOZ) to direct airflow toward the cows. Although previous studies have suggested that baffles can substantially amplify the system's cooling effect, the comprehensive impact of baffles on airflow patterns in a full-scale barn is less understood. Traditional measurement techniques, involving physical sensors, are both technically demanding and costly. Moreover, they often fall short in accounting for the dynamic microenvironmental changes induced by fluctuating weather, animal movement, and operational machinery. This study leverages computational fluid dynamics (CFD) to model airflow behaviors within a cross-ventilated barn, specifically examining the influence of a retractable baffle. CFD not only offers a reliable and efficient method for simulations but also allows for accurate assessments by validating the generated data against empirical observations. The results verify that, when properly configured, air baffles can significantly enhance airflow at cows in large barns. Additionally, the study establishes the reliability of CFD for designing large-scale dairy barns.

**Keywords:** dairy cows; retractable baffle; heat stress; airflow patterns; ventilation



## 1. Introduction

Heat stress has become a significant challenge for the dairy industry, not only because climate change has produced longer and hotter summers with frequent and extreme heat waves but also because modern dairy cows produce substantially more milk than their predecessors. Mechanical ventilation systems have been widely adopted by the U.S. dairy industry to effectively mitigate the presence of heat stress, which negatively impacts both the welfare and the productivity of dairy cows [1]. The cross- and tunnel-ventilation configurations are the preferred structures when designing a barn that is mechanically ventilated [2–4]. These systems employ a series of fans. However, forced airflow through the barn does not necessarily guarantee an improved convective cooling effect. According to [2,5], to achieve an adequate cooling effect, the airflow velocity should be at least 1 m/s in the animal-occupied zone (AOZ), which is where the cows spend most of their time. However, achieving and maintaining the cooling effect is not a simple matter, because air tends to flow along the least-resistant path, particularly through the open space above and the alleyways that run between the rows of stalls in the AOZ. Consequently, a significant amount of the energy used by the fans is wasted, the flow of air through the AOZ being considerably less than the rate as measured at the fans. Morden dairies are now building barns that house more cows (often 500 or more), with each cow generating more heat energy than in the past [6]. For these reasons, various studies have sought to measure the effectiveness and the importance of the mechanical ventilation system typically used in

dairy barn designs. Moreover, as a result of their findings, these and other researchers are calling for additional work aimed at improving the efficiency of mechanical ventilation systems, which can be resource-intensive and therefore unduly costly, and such inefficiency will greatly influence the long-term viability of the U.S. dairy industry [7].

Air baffles have been installed in many dairies in order to improve the "cow-side" cooling effect produced by the barns' mechanical ventilation systems. The baffles, which are suspended from the ceiling and hang parallel to the inlet openings, redirect the fan-driven airflow and increase its velocity through the AOZ [5]. Harner and Smith [8] found that installing baffles increased the air speed in the freestall area from a range of 0.89 to 1.34 m/s up to a range of 2.68 to 3.58 m/s. The low-profile cross-ventilation (LPCV)-type barns are widely used for large-scale barns (>500 cow heads) because their wide width, relative to their low height, provides the baffles with the most effective profile for accelerating the airflow in the AOZ [3]. It is generally agreed that baffles can significantly improve the cooling effect produced by mechanical ventilation systems. However, accurately measuring the influence that baffles exert on airflow patterns is extremely difficult due to the sheer scale of dairy barns, the topological complexities created by dynamic structural uncertainties (such as cow movements), and the heterogeneous distribution of the wind speed and air temperature inside a barn. Also, the traditional ways of using physical wind sensors to assess indoor airflow are labor-intensive, time-consuming, and costly. Furthermore, because each dairy barn design is unique, a method that allows a rapid and accurate assessment of the airflow inside different buildings is critical. Computer-generated, model-based computational fluid dynamics (CFD) could significantly reduce the cost and the time required to make an assessment of the internal structures (such as baffles) by accurately simulating the airflow patterns throughout the barn. Once properly validated, the results of the CFD simulations could be used not only to objectively assess the influence that baffles have on an indoor airflow but also to reliably predict any changes in airflow behavior that may occur as a result of a proposed structural modification (i.e., the installation of additional structures) and, thus, could significantly reduce both the amount of labor involved in making the change and the economic cost.

The CFD approach has been used to generate airflow simulations involving a wide variety of animal houses. Wu et al. [9] used a CFD-based approach to assess various methods for determining the air exchange rate in a naturally ventilated dairy barn operated under conditions associated with natural ventilation. Rojano et al. [10,11] also utilized computer modeling done with CFD to determine the impact that air discharged from a poultry house may have on the surrounding environment. In a more recent study of naturally ventilated animal housing, Tomasello et al. [12] were able to develop a CFD model that could simulate natural ventilation in a semi-open freestall barn designed to house dairy cows, and Mondaca et al. [2] demonstrated the benefits of using CFD to assess the microenvironments that develop within tunnel-ventilated dairy facilities. Additionally, Zhou et al. [3] used CFD to successfully model a sectional cross-ventilation barn and investigate the optimal baffle placement in relation to the heat transfer rate on cow body surfaces. Many other studies have also been conducted in order to improve the CFD method as it applies to creating animal housing models. Mondaca and Choi [13], for example, evaluated different modeling approaches with respect to their efficacy in terms of decreasing the computational cost of using a simplified cow geometry, as well as a porous medium, to simulate animals in the AOZ. Besides demonstrating the CFD's ability to simulate airflows, researchers have also presented a series of simulation outcomes for the gas emissions and heat and mass transfers that occur inside livestock houses [14–16]. Moreover, Rong et al. [17], in an effort to help other researchers use CFD to model livestock housing, summarized the best practices associated with CFD modeling, specifically with respect to simulating livestock buildings whenever selecting an appropriate turbulent-flow field model and numerical solver scheme was at issue. Additional information concerning the use of CFD to predict general indoor airflow can also be found in well-documented manuscripts summarized by Sørensen and Nielsen [18] and Nielsen [19].

This study aims to simulate airflow patterns in a large-scale dairy barn, and in doing so, it also assesses the impacts of baffle structures on the airflow pattern and how that impact relates to cow cooling in the AOZ. Two different cases were analyzed, both involving the same model of a commercial cross-ventilated barn, which is commonly used in Wisconsin, USA. The physically measured data from a barn and the simulation predictions of the airflow from the corresponding CFD model to the barn were then compared in order to validate the CFD outcomes against the experimental data. A porous media model of cows was also implemented to achieve a better understanding of flow patterns occurring at a key location in the barn where the geometric difference between the two cases (with respect to the baffle structures) was the most significant.

## 2. Materials and Methods

### 2.1. Experimental Site and Barn Design

A commercial dairy farm in Watertown, Wisconsin, USA, was selected as the experimental site for this study. As shown in Figure 1, at the time of the study, the dairy farm had four naturally ventilated barns and one mechanically ventilated barn (that served as the study's experimental site). Most of the other buildings on the farm were open structures. The mechanically ventilated barn's inlet openings were located at the north end of the facility, and the outlets were oriented south. The facility, which was characterized as a low-profile cross-ventilated (LPCV) barn, was located at the south end of the farm and was directly connected to the milking parlor, along with one of the naturally ventilated barns. The farm was surrounded by flat, open areas featuring scattered obstructions that created wind exposure. Due to the relatively flat land surrounding the site and the comparatively low height of the obstacles in the neighborhood of the studied cross-ventilated barn, this study did not take into consideration any wind–shade effects that may have been produced by the surrounding obstacles or land topology.

The side and isometric views of the cross-ventilation dairy barn are shown in Figure 2. The barn was 81 m wide and 62 m long in the main airflow direction. The ridge height of the facility was 6.17 m. There were previously installed four fixed baffles and one newly installed retractable baffle arrayed along the ridge in the middle of the barn. That is, the producer noted the lack of adequate wind speed for cows along the middle rows and decided to add an additional retractable baffle. This type of expensive and time-consuming trial-and-error redesigning and retrofitting is quite common in real-world situations, generally without little help from design engineers with adequate backgrounds, such as in CFD, experimental measurements, and data analyses. The retractable baffle was either fully retracted or extended throughout testing in the two cases studied in this paper. Compared to inexpensive fixed baffles, the advantages of the retractable baffle are (i) to improve the airflow at lower ventilation rates and (ii) to prevent the trapping of stale air between the baffles when fast air speeds are not needed to mitigate heat stress. The bottom edges of the fixed baffles were located at a height 2 m above the sand beds, extended to 0.2 m above the floor, and are marked with L-shaped blocks in the figures. When fully released, the bottom edge of the retractable baffle extended downward 0.2 m further than the bottom edges of the fixed baffles. Figure 2a,c depict the barn with the retractable baffle fully extended, whereas Figure 2b depicts the barn when the retractable baffle was fully retracted. The barn had two 5.13-m-wide drive-through feed lanes running across from east to west (each lane having a retractable door at either end) and one transfer lane running through the center of the barn from north to south. More detailed views and dimensions of the barn are shown in Figure 2d with detailed dimensions in A-G. During the study, the airflow was driven by a negative pressure generated by a series of 37 small and 8 large exhaust fans mounted on the outlet wall at the south end of the barn. Along with the inlet openings on the north side of the barn, the transfer lane, which, at the time of the study, ran across the middle of the barn and toward the milking parlor, was kept open to act as another inlet. The side doors connected to the feed lanes were opened periodically in accordance with operations related to feed distribution.

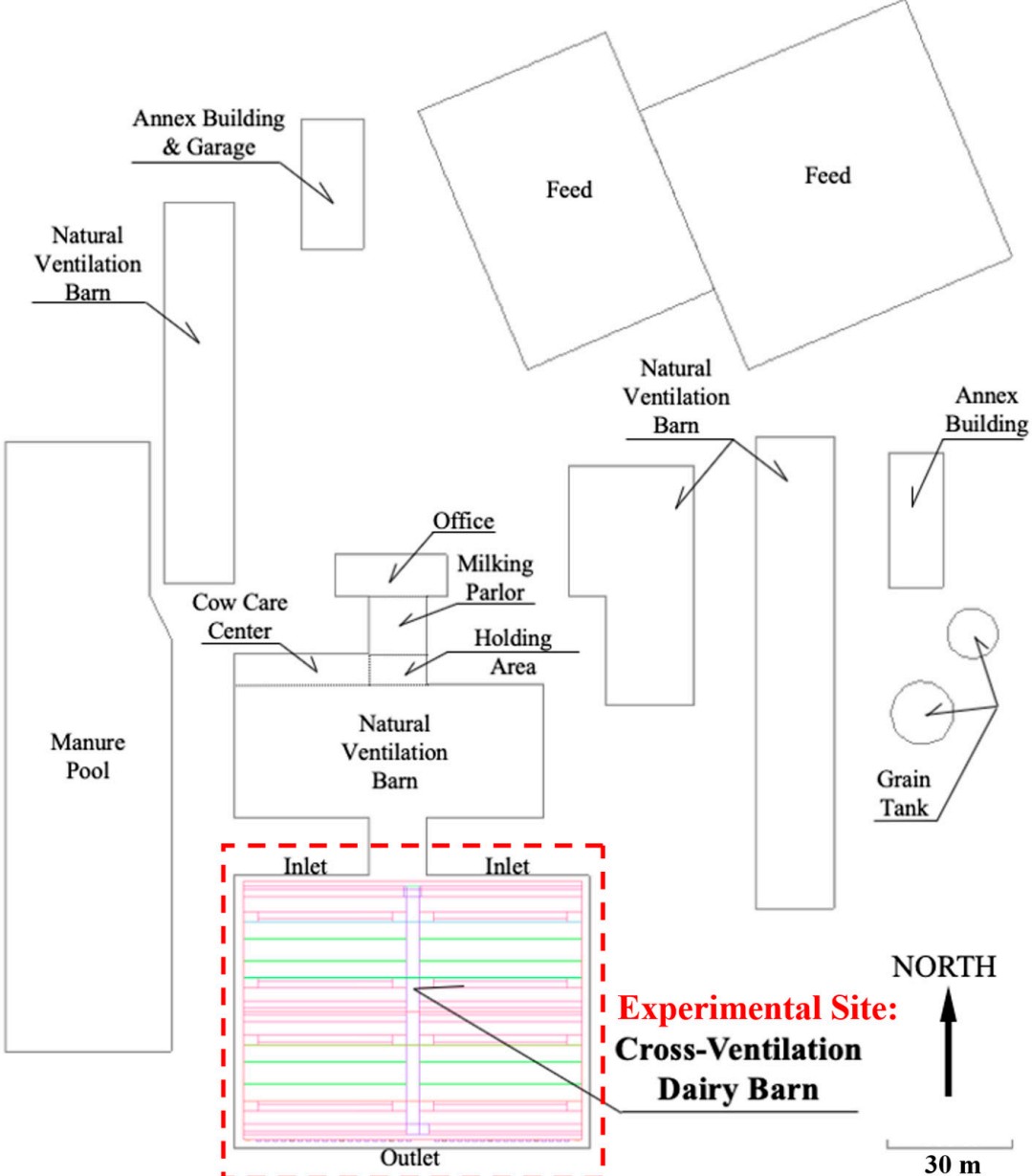

**Figure 1.** A plan view of the experimental dairy farm. The low-profile cross-ventilation dairy barn under study was located at the south end of the farm.

The northeast inlet's opening measured 37.73 m², and the northwest inlet opening measured 33.46 m². Additionally, the transfer door to the milking parlor was opened 9.64 m². When fully occupied, the barn could house approximately 600 adult Holstein dairy cows in 8 rows of freestalls. The barn was divided into 8 separate sections of tail-to-tail pens, with each section able to accommodate 72 to 76 cows (except one section that could accommodate 54 cows).

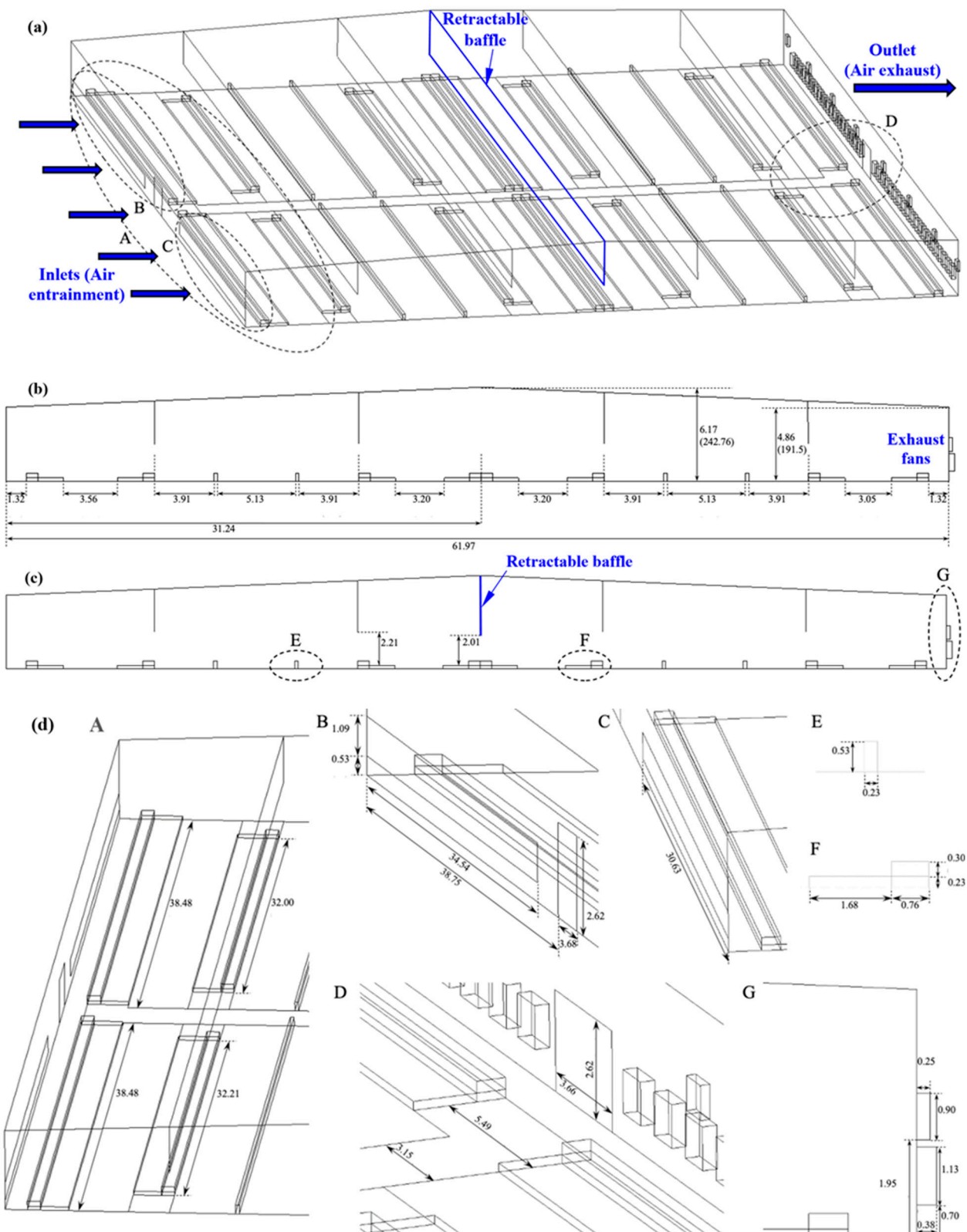

**Figure 2.** Isometric (**a**) and side (**b**,**c**) views of the cross-ventilated barn with respect to both the four-baffle and the five-baffle cases (including the retractable baffle). Detailed views of the facility, including the dimensions, are shown in (**d**). Detailed dimensions are shown in A–G.

### 2.2. Experimental Data

2.2.1. Data Measuring Tools

A pair of wind velocity sensors (accuracy to within ±3% or ±0.015 m/s, Alnor Velometer thermal anemometer AVM440, TSI Inc., Shoreview, MN, USA) were used to compile data pertaining to the airflow speed that occurred inside the cross-ventilated barn and at the inlets and the outlets during the test period. All the airflow data were collected at 1-s intervals over a period of 3 min at each measurement location. To best represent a resting cow and a standing cow, one sensor was attached to a tripod at a height of 0.5 m above the sand bed, while the other was set at 1.5 m.

2.2.2. Timeline and Measurement Locations

A number of visits to and preliminary CFD simulations of the experimental cross-ventilated barn were conducted using ANSYS-Fluent (release 2019.R2, ANSYS, Canonsburg, PA, USA) in order to pinpoint the optimal measurement locations that could accurately represent the overall airflow patterns occurring inside the barn. Both the preliminary tests and the experimental data measurements were conducted during the summer (from July to August of 2019) when the weather was hot and humid and the fans were operating at full capacity. Following the preliminary CFD tests, the airflow pattern data compiled were analyzed in order to identify the best measurement locations (that are shown in Figure 3, along with the location coordinates). The blue box zones shown in the figure represent the sand bed freestalls, the green box zones designate the drive-through feed lanes, and the red lines designate the baffle locations. A total of 45 nodes (represented by solid and shaded blue dots) were chosen as the measurement locations. For the blue solid dot locations, the rows are labeled with numbers and the columns are labeled with letters. For all the blue dot locations, both wind sensors, set at heights 1.5 m and 0.5 m above the ground surface, were used to measure the velocity, except at the nine locations along rows 1 and 9, where only the sensor at a height of 1.5 m was used to obtain the velocities at the inlets and the outlets. Four shaded blue dot locations, labeled H, I, J, and K, were placed directly underneath the installed retractable baffle so that the extended baffle's influence on the velocity profile could be compared to the profile when the baffle was retracted. Due to the inconsistency in sand bedding topology, the bottom edge of the extended retractable baffle was used as a reference point when determining the six profile heights to the floor surface (including the sand bed), which were set at 1.45, 1.20, 0.95, 0.70, 0.45, and 0.20 m.

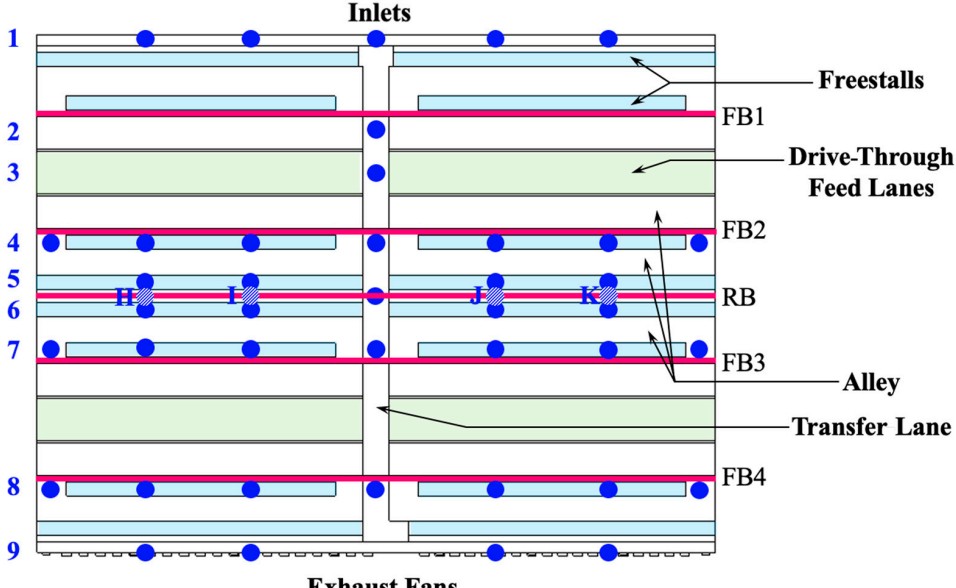

**Figure 3.** Data measurement locations for field data in the cross-ventilated barn. The numbers on the left (1–9) denote the row numbers.

The field measurements were carried out for six days (from DOY 227 to 233). On each day, the field data were collected at all 45 measurement locations. In each study case, field measurements were conducted for three days to produce triplicate readings, giving a total of 540 samples at each location per study case. DOY 227, 228, and 231 were the days on which the retractable baffle was fully extended, and DOY 228, 230, and 233 were the days on which the retractable baffle was fully retracted. No data was collected on DOY231, because the sand beds were cleaned and sand replenished. All the data were collected between 9 a.m. and 5 p.m. on each day. During the measurement period, no rain or fogging events occurred, and the weather patterns, in terms of air temperature and humidity, remained nearly the same.

*2.3. Computational Fluid Dynamics Models*

The virtual representations of the two study barn cases (both relying on the retractable baffle extension) were modeled using CFD. As shown in Figure 2, one case involved a four-baffle barn equipped with a fully retracted baffle, and the other case involved a five-baffle barn equipped with a fully extended baffle. A full list of the boundary conditions used in the simulations is shown in Table 1. To emulate the negative pressure-driven exhaust fans, the fan outlets were set as the velocity inlet conditions with a negative value averaged from the field data. To simplify the calculations, both large and small fans were assumed to produce the same velocity magnitude. The inlets were set as pressure outlets with zero gauge pressure, and the transfer lane door was given as a positive inlet velocity, which was the average velocity entering from the adjacent naturally ventilated barn and from the milking parlor. All the geometric surfaces in the computational domain were set as no slip walls, and for simplicity, the mass transfer occurring as a result of the manure and water present in the barn was not considered.

**Table 1.** Boundary conditions for the CFD simulations based on the field data.

| Locations | Types | Boundary Conditions |
|---|---|---|
| Inlets | Pressure Outlet | Turbulent Intensity: 5% <br> Gauge Pressure: 0 Pascal |
| Transfer Lane Door | Velocity Inlet | 2.77 m/s (without Retractable Baffle) <br> 1.99 m/s (with Retractable Baffle) |
| Outlets (Exhaust Fans) | Velocity Inlet | −4.41 m/s (without Retractable Baffle) <br> −4.15 m/s (with Retractable Baffle) |
| Sidewalls, Ceiling, Floor, Freestalls, and Other surfaces | No slip | Wall |

A porous model of a typical cow was used in order to consider the influence that the cows exerted on the velocity profile that occurred beneath the retractable baffle. Porous blocks were placed along row 4 in Figure 3. A porous block representation that combined the two basic cow postures was considered as a way to simplify the CFD geometric meshing and to reduce the computational cost, all while maintaining results comparable to those produced by the simulation that used real cow models. The porous cow model used was developed and implemented during this study by applying the method presented in Mondaca and Choi [13] and Drewry et al. [16]. To calculate the viscous and inertial resistances of a non-Darcian porous cow model, a single row of cow geometries (3 standing and 9 lying) was used in accordance with the study by Overton et al. [20], which found that, on average, about 75% of the cows inside a dairy barn at any one time are lying down. Also of note, the calculations were performed with velocities ranging from 0.5 m/s to 3.6 m/s, which is a typical velocity range inside a dairy barn during hot and cold seasons [4,21,22].

For the CFD simulations, the mesh grid used throughout the computational domain was discretized with a structured hexahedral grid. A relatively coarse mesh density was maintained throughout the domain to reduce the computational burden. However, finer meshes were applied under the baffles and on the wall surfaces to accurately predict

redirected and accelerated airflows. As recommended by Rong et al. [17], for both computational models, adequate Y-plus value ranges of around 76 were obtained on the surfaces of the sand bed geometries, and a high overall mesh quality of 0.95 was obtained in each case. A series of mesh tests was conducted by changing the total number of cells until there were no significant changes in the overall shear stresses exerted on the freestalls. The final total number of discretized cells in both CFD computer models was around 96.4 million.

For the CFD simulations, the steady-state Reynolds-averaged method for solving the Navier–Stokes equations (RANS) was applied using a finite volume approach. The realizable *k-ε* model was used to represent the turbulence flow field in the simulation. The enhanced wall treatment function was applied in conjunction with the turbulence equation to simulate the real airflows occurring in the facility. The realizable *k-ε* model can predict air movements inside buildings by applying improved equations for determining the rate at which turbulent kinetic energy dissipates and can predict the rotation, recirculation, planar and round jets, and the boundary layer as well under a strong adverse pressure gradient when compared to the preceding model by Menter [23], which is the standard *k-ε* model. The RANS model was solved using the SIMPLE (Semi-Implicit Method for Pressure-Linked Equations) numerical scheme and the second-order upwind differencing scheme (UDS) due to their reliability as proven in other relevant studies [2,3,17,24]. The buoyant force was neglected, since this effect is generally negligible in both forced convective and naturally ventilated dairy barns unless the incoming wind velocity is insignificant [25,26]. Double-precision variables were used in all the simulation calculations, and all the cases were iterated until the following convergence criteria for the CFD simulations were achieved: the residual of the net mass flow rate in the computational domain was less than $10^{-6}$, the residuals of velocity and pressure were less than $10^{-6}$, and the residual of the area weighted-averaged shear stress on the freestalls was less than $10^{-6}$.

### 3. Results and Discussion

#### 3.1. Field Data Measurements

In both cases, all the wind velocity data (in m/s) obtained at each of the measurement locations inside the cross-ventilated barn, as presented in Figure 3, were averaged and are shown in Figure 4. Except for the inlets and outlets, all the locations in each case showed two values with the standard deviation indicated inside parentheses. The top value represents the velocity measured at a height of 0.5 m, while the bottom value represents the velocity measured at 1.5 m above the ground surface. In both study cases, a consistently higher velocity was recorded in the transfer lane at all lateral points, except at the inlet transfer door. This is likely because the transfer lane offered an open pathway through the barn and therefore considerably lower air resistance compared to the other sections of the barn.

Overall, the measurement points located downstream from a baffle indicated higher velocities when compared to the points located upstream from a baffle and met the recommended minimum cooling air speed of 1 m/s [5]. Between the two study cases, significant increases in velocities were observed, especially at the locations along rows 5, 6, and 7 (see Figure 3 for the row numbers) and similar to those obtained by Harner and Smith [8]. The wind velocities at most locations roughly doubled. However, some locations captured a velocity increase of as much as six times faster. Especially along rows 6 and 7, the presence of a retractable baffle improved the wind velocities to well over the minimum cooling air speed; otherwise, the velocity through these stalls barely met the minimum requirement. Slight decreases in the velocities at the outlets were also observed when the retractable baffle was fully extended. This was potentially due to the additional air resistance generated along the dominant airflow stream, which was imposed by the fully extended retractable baffle (that narrowed the cross-sectional area of the flow path through the barn). The presence of the retractable baffle also increased the overall wind velocity along the transfer lane by about 10%.

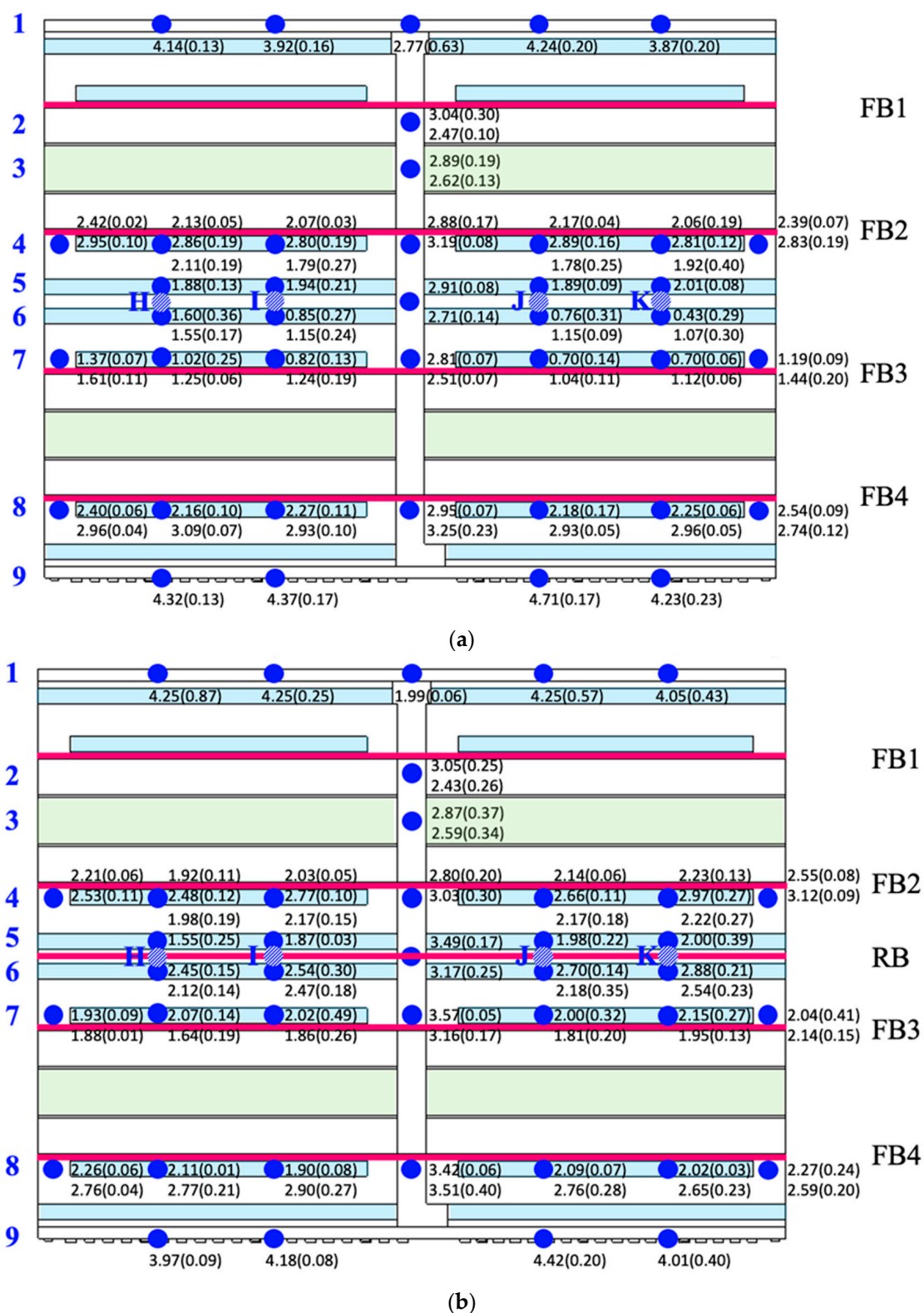

**Figure 4.** Measured wind velocity data with standard deviations in parentheses for each measurement location for both study cases: (**a**) retracted and (**b**) extended. The four fixed baffles are indicated as FB1, FB2, FB3, and FB4, and the retractable baffle is indicated as RB. The numbers on the left (1–9) denote the row numbers.

Certainly, the field data obtained from the experimental site should give clear indications of how the presence of the retractable baffle affects the airflow through a cross-

ventilated dairy barn. However, the procedure used to obtain field measurements in a barn is extremely time- and labor-intensive, and it is almost impossible to fully understand how the actual governing airflow pattern behaves and to capture developed airflow patterns by means of physical measuring tools, especially in between the measurement locations.

*3.2. CFD Outcomes*

3.2.1. Overall Airflow Pattern in a Dairy Barn

The CFD simulations created in this study were based on the boundary conditions defined by the measured data. The isometric views of the velocity contours and vectors in both cases are shown in Figure 5. In mechanically ventilated dairy barns, airflows tend to follow the path of least resistance [4]. The transfer lane, which offers a mostly open path through the center of the barn, impacts the airflow patterns that develop in different sections of the barn (depending on the section's proximity to the transfer lane). For example, as Figure 5b (between "A" and "B") shows, significantly different (and reduced) velocity contours occur, especially along rows 5, 6, and 7, and as a result, a symmetric distribution of airflow across the barn was captured in reference to the transfer lane. It should be noted that the size and the location of the transfer lane can greatly influence the distribution of airflow through the barn.

The velocity contours and vectors that developed in both cases (four-baffle and five-baffle scenarios) are depicted (in elevation view) in Figure 6. The elevation views in Figure 6 are of perpendicular sections of the inlet walls, as indicated by the cross-section "A" in Figure 5. The overall velocity contours in both cases were similar to those obtained by Zhou et al. [3] with respect to a sectional simulation of a large-scale cross-ventilated dairy barn equipped with baffles. Specifically, the baffles significantly accelerated and redirected the airflow, channeling it closer to the floor and through the AOZ, which is where the cows reside most of the time. With respect to the four-baffle case shown in Figure 6a, the air passing between the middle two baffles (FB2 to FB3) flowed at a low velocity, possibly because the cross-sectional area was larger and longer between these baffles. Because more than 50% of the cows inside an AOZ will typically reside in between these two baffles, a relatively lower airflow speed (compared to the velocities occurring in the other sections of the barn) can produce a significantly negative impact on the dairy farm's total production, a majority of the cows having been insufficiently cooled and the system's fans having operated at an inefficient level. When the retractable baffle was extended (to create the five-baffle case, as shown in Figure 6b), the velocity of the airflows passing through the four rows of stalls in the middle section significantly increased. Moreover, the airflow patterns that developed closer to the inlets and the outlets tended to remain consistent, regardless of whether the retractable baffle was present or not.

To better understand the airflow paths that occurred in both study cases, particle tracking was performed, as shown in Figure 7. A total of 56 particles were released at the inlets (two curtain openings and the transfer door) that were equidistant from each other. As the figure shows, the baffles produced a constant acceleration and redirection of the airflow. While the velocity constantly changed throughout the freestall zones, the airspeed remained relatively constant through the transfer lane. A comparison of the two cases showed that the presence of a retractable baffle effectively redirected the airflow to enhance the velocity magnitudes at the cow level. The air coming in through the inlets flowed along a relatively straight path throughout the first half of the barn; however, in both cases, the air particles revealed that backflows occurred closer to the outlet wall, no doubt due to the sudden decrease in the flow area at the outlets. In general, the upstream flow patterns occurring inside the barn were relatively symmetrical because of the geometrical symmetry of the facility in relation to the transfer lane right in the middle.

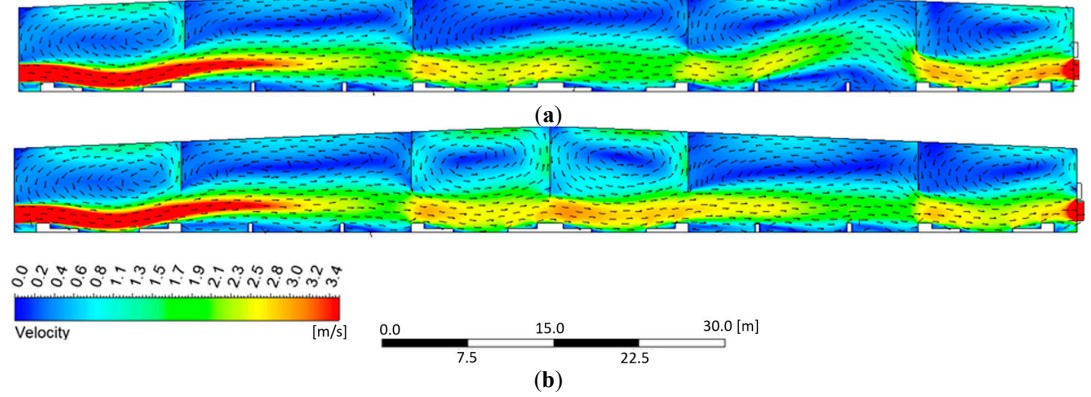

**Figure 5.** Isometric views of wind velocity contours and vectors for both study cases: (**a**) retracted and (**b**) extended.

**Figure 6.** Side views of the wind velocity contours and vectors for both study cases: (**a**) retracted and (**b**) extended.

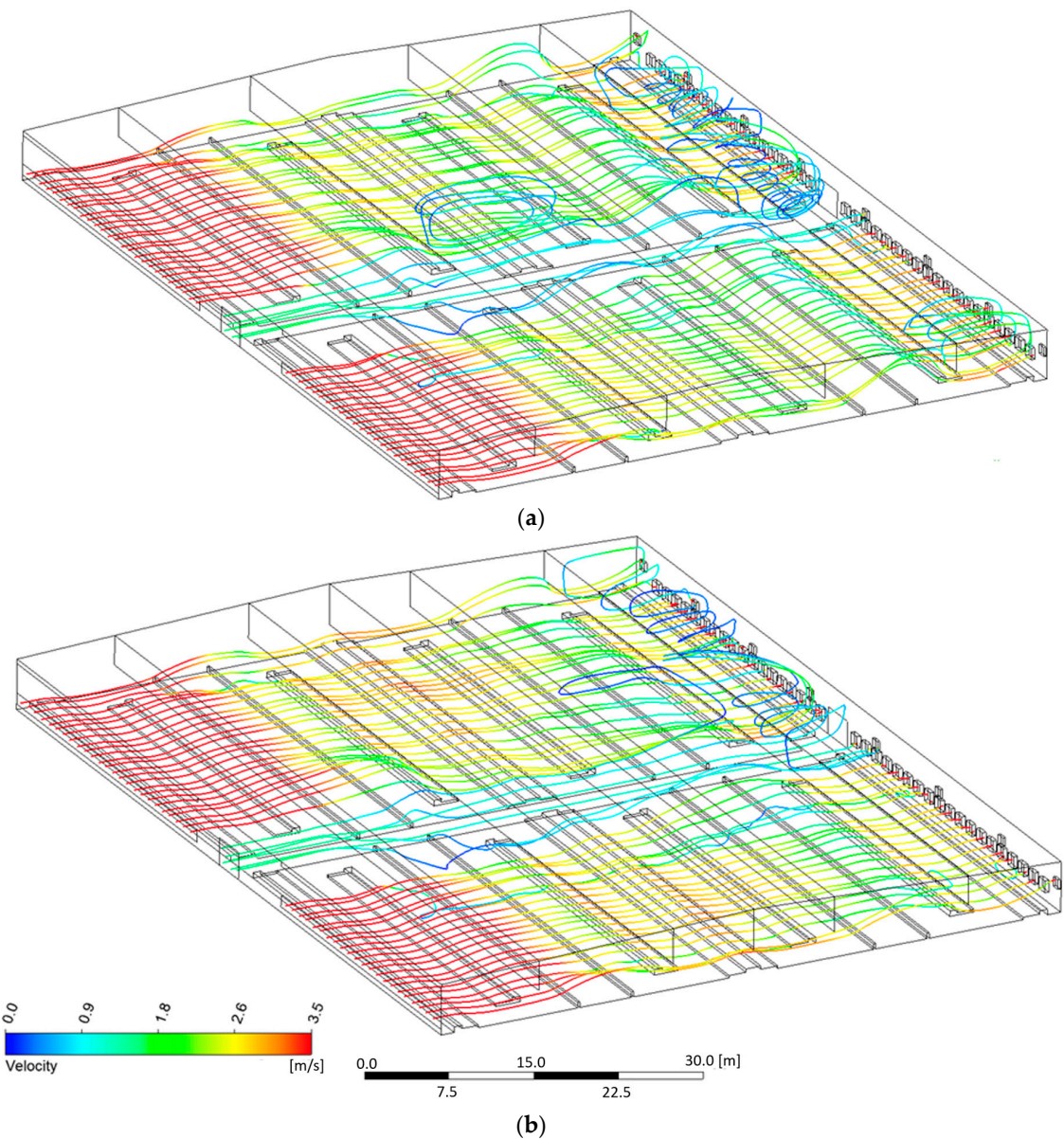

**Figure 7.** Lagrangian particle tracking simulations—airflow pattern particle paths through the entire barn for both cases based on 56 released particles from the inlets: (**a**) retracted and (**b**) extended.

### 3.2.2. Application of the Porous Cow Model

Figure 8 depicts the velocity contours near the retractable baffle location in both cases when the porous cow model is present. As shown in the figure, the porous cow models were placed on the sand beds along row 4. Owing to the presence of the additional air resistances produced by the porous cow models, the airflow patterns that developed at decelerated wind velocities were found to be similar to those shown in Figure 5.

Figure 9 shows the velocity contours (in a cross-sectional view in both cases) that occurred under the ridgeline of the barn, which are also shown in Figure 8a,b. The presence of the retractable baffle dramatically enhanced the overall wind velocity magnitudes below the ridgeline. Some of the freestalls closer to the transfer lane, however, did not register improvements in the wind velocities that were as pronounced as those recorded in the other freestalls. This is likely due to the adjacent open space (with a lower air resistance path) that runs along the transfer lane and allows air to escape from the path through the freestall zones. Here, CFD was able to capture a nonuniformity in the velocities occurring beneath the baffle, a phenomenon that would otherwise be extremely challenging to investigate

using physical tools. Furthermore, unlike the measurements acquired via physical field methods, CFD-generated data can predict wind velocities as the field points not only those locations but also throughout the entire barn.

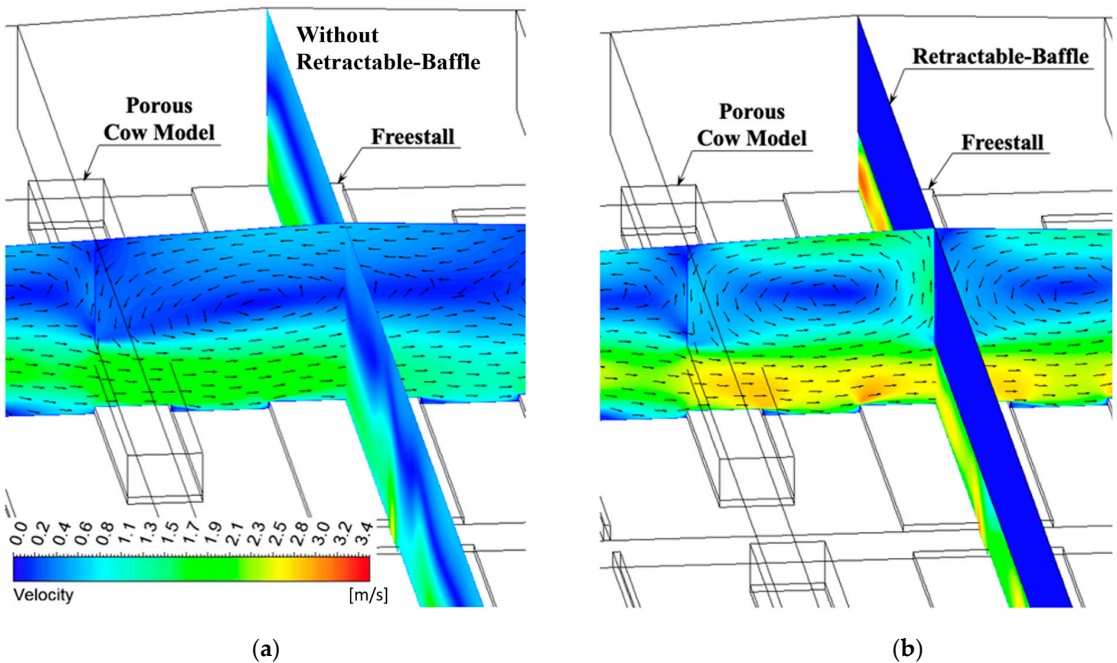

**Figure 8.** Wind velocity contours and vectors for both study cases, without (**a**) and with (**b**) the retractable baffle, with porous cow blocks at row 4.

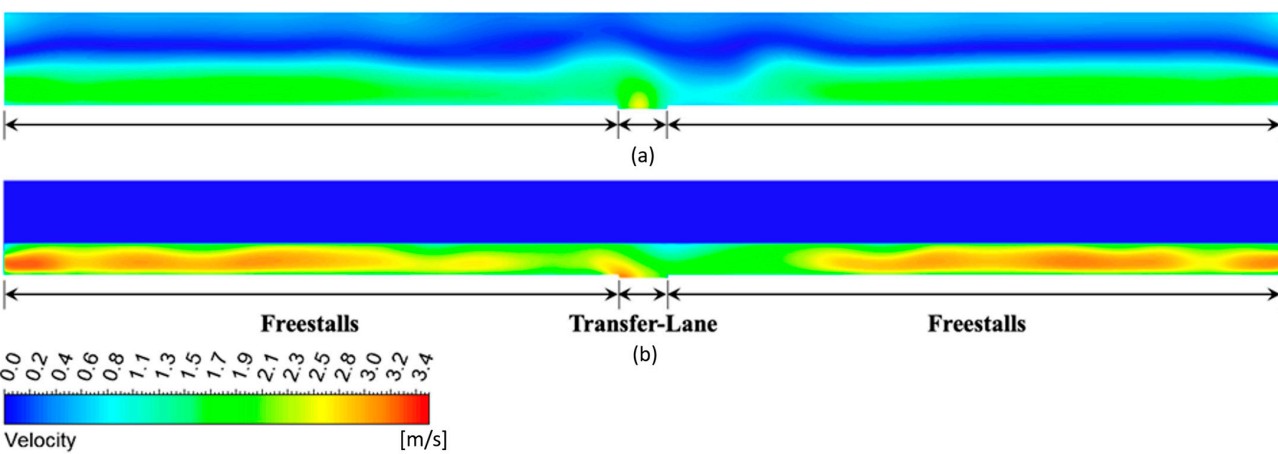

**Figure 9.** Wind velocity contours along the cross-sections without (**a**) and with (**b**) the retractable baffle.

*3.3. CFD and Field Measurement Outcomes Comparison*

3.3.1. Velocity Profile Comparisons

Figure 10a,b show, for each study case, the velocity profile comparisons between the field data obtained at the four locations (H, I, J, and K) and the CFD outcomes. The Y-axis indicates the distance from the bottom edge of the retractable baffle toward the sand bed surface. The heights (the back of a lying cow and that of a standing cow) are also indicated in the figure. For each case, the CFD predictions associated with the four different locations were averaged out and plotted along a single line, whereas, for the comparisons, the field data were plotted individually based on the four different locations. When the retractable baffle was not present, each profile obtained from the field data showed a trend that varied from the others and did not develop any particular singular profile pattern.

Conversely, when the retractable baffle was present, the velocity profiles at each location converged into profile patterns that were similar. This is likely because the presence of the retractable baffle significantly tightens the cross-sectional area available for airflow, which compresses the airflow path and mitigates the flow fluctuation. As a result, the overall average velocity magnitude increased by approximately two-fold, with the highest wind velocity captured at a height of 0.70 m beneath the baffle. The CFD simulations showed that the highest velocity was captured at a height of 0.95 m beneath the baffle when the baffle was present. The CFD model that included the porous cow model predicted an overall similar range of wind velocities at the H, I, J, and K locations when compared to the field measurements. Accurately capturing the actual flow field using CFD at such a scale is challenging, because many factors influence the entire airflow occurring in the barn and also because the assumptions used in flow field models and in the computational geometries created for the CFD simulations were simplified versions of those factors. Nevertheless, the CFD models used in this study were able to capture velocity profiles comparable to those derived from the field data, especially at the lying cow zone, which is the most important target zone in terms of cow cooling in the AOZ. Because dairy cows spend most of their time lying down on the sand bed [27–30], a comparison of the wind velocity through the lying cow zone is crucial in a dairy barn design aimed at providing the animals with an adequate and consistent cooling effect.

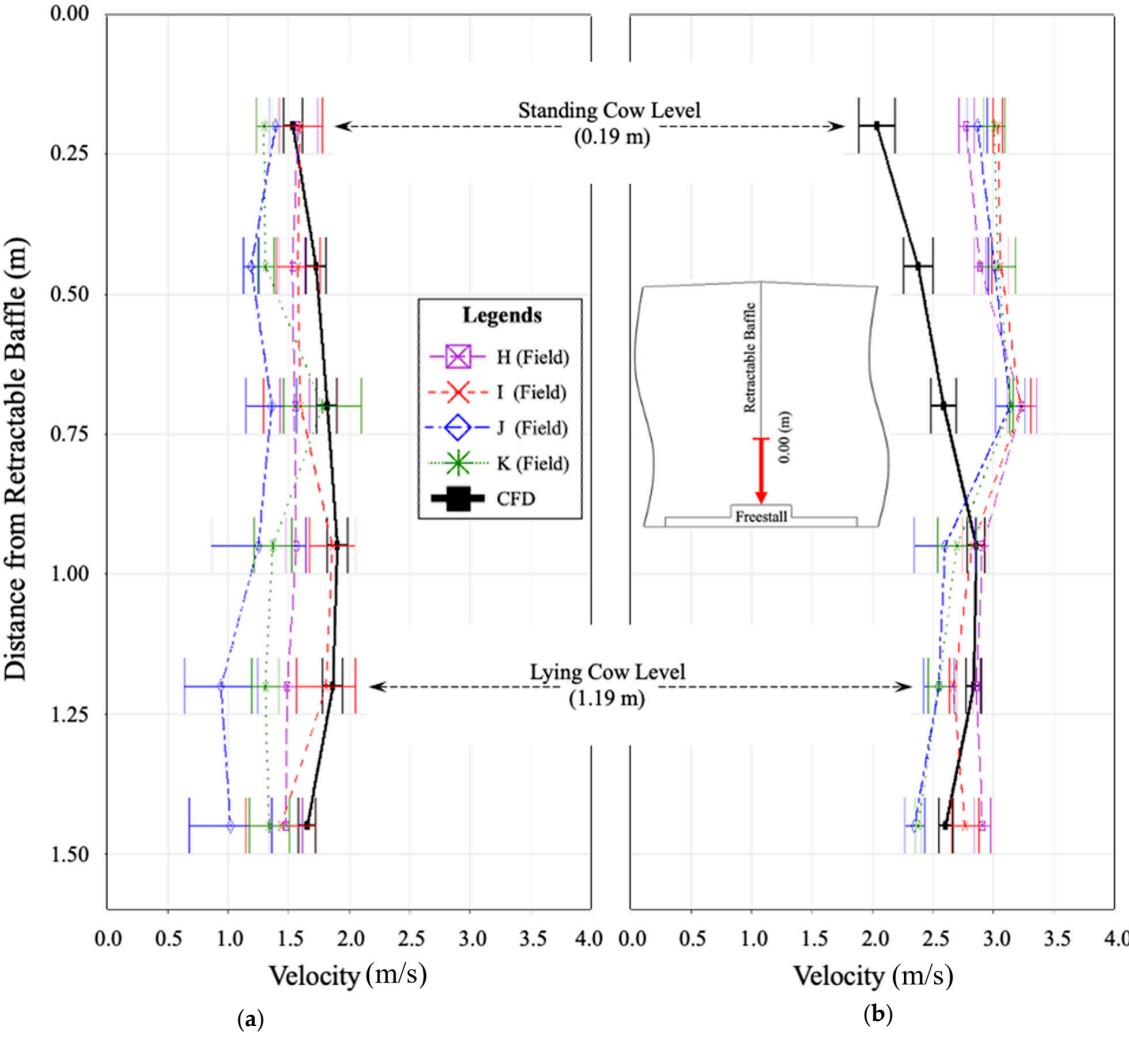

**Figure 10.** Comparisons of wind velocity profiles between measured data at each location (H, I, J, and K), and the averaged CFD prediction from the corresponding locations for both cases. The left (**a**) and right (**b**) figures indicate the cases without and with the retractable baffle, respectively.

### 3.3.2. Overall Velocity Comparisons

The CFD outcomes were compared to the field data compiled at rows 4, 5, 6, 7, and 8 at both heights of 0.5 m and the 1.5 m. As can be seen in Figure 11, the measured data pertaining to each row were combined and represented in the box plots with maximum and minimum values for the two different study cases, along with the first and third quartiles and medians. Almost no skewness was observed in the distribution of any of the measured wind data samples. The CFD outcomes obtained from the corresponding data measurement locations in each row were also averaged and plotted in the same figure. Overall, the CFD results showed good agreement with the averaged measured data, and both were able to capture the changes in the flow trends occurring in each case, with and without the retractable baffle. The presence of the retractable baffle significantly increased the air velocities at both heights 0.5 m and 1.5 m across rows 6 and 7 (that are located directly downstream of the retractable baffle), and hence, all the locations received a cooling air velocity that was above the recommended minimum velocity of 1 m/s [5]. Such was not the case for rows 6 and 7 when the baffle was not extended.

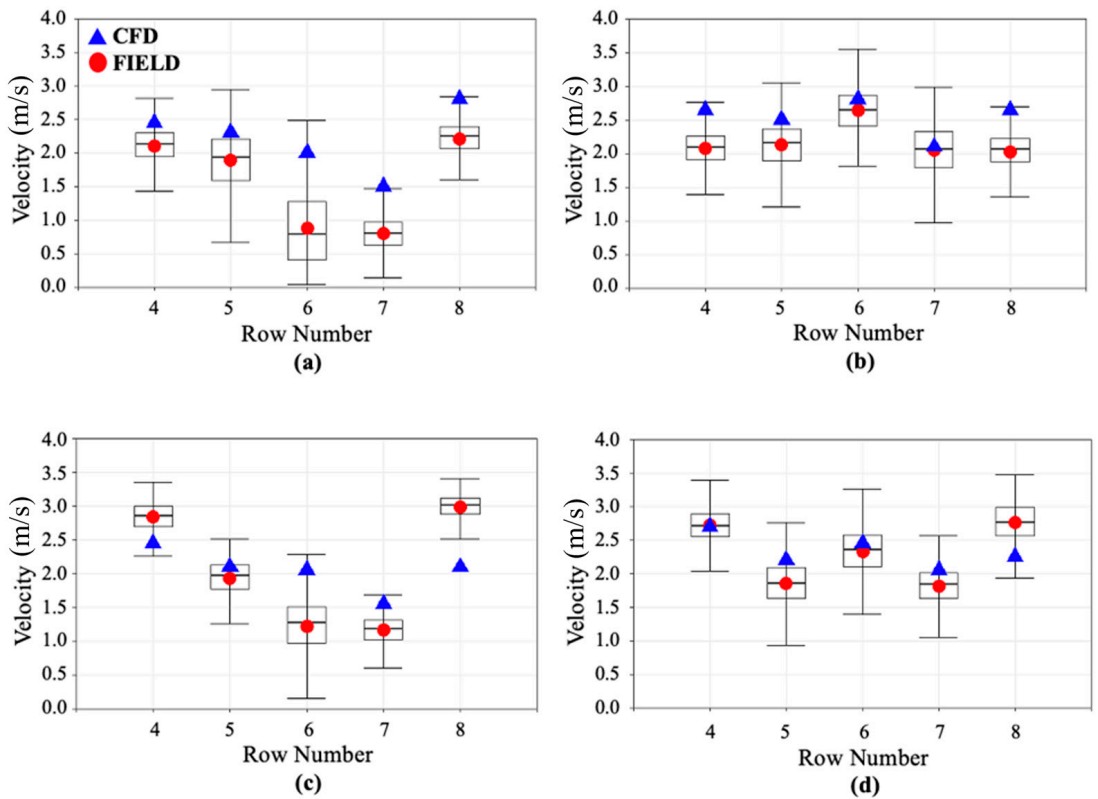

**Figure 11.** Wind velocities at the lying cow ((**a**,**b**), 0.5 m) and standing cow levels ((**c**,**d**), 1.5 m) without (**a**,**c**) and with (**b**,**d**) the retractable baffle.

Being supplied with the measured boundary conditions and corresponding computational barn geometries, the CFD models developed in this study proved capable of producing reliable replications of the flow fields inside a large-scale cross-ventilated dairy barn equipped with multiple baffle structures designed to redirect airflow. The CFD simulation provided a comprehensive understanding of the influence that these baffles exerted on the airflow patterns occurring in the cross-ventilated barn used in the study and, in so doing, showed that baffles can significantly increase the wind velocity in the AOZ at the cow level and, in turn, increase the convective heat removed from the cows by the airflow. The data collected from the barn that originally lacked the retractable baffle showed that, once the retractable baffle was added to the ridgeline at the proper location in the middle of the barn, the overall airflow improved. Moreover, when these CFD outcomes were

compared to the experimental data, the predictions were found to be valid, and it could be concluded that not only can CFD accurately assess the effects that baffles can exert on airflows occurring in existing structures but also serve as a source of referencing data that should help designers and engineers plan structures that have not yet been built.

## 4. Conclusions

This study involved comparing data acquired in the field to be generated by means of computational modeling in order to assess the extent to which air baffles may influence the flow patterns that develop as air passes through a cross-ventilated dairy barn inside a large dairy barn. The field data indicated that installing baffles at key locations in cross-ventilation dairy barns will significantly improve the convective cooling performance of the ventilation system by accelerating and effectively redirecting the dominant airflow downward into the AOZ. When compared to the field data, the data generated by the CFD simulations were found to reliably predict the flow fields in both study cases (that involved different numbers of baffles in barns of identical dimensions); specifically, both sets of data showed good agreement in their velocity magnitudes and profiles, especially those occurring near the retractable baffle.

Given that the traditional way of investigating airflow patterns inside a barn is extremely labor-intensive and time-consuming, the numerical approach this study used, which included an application of porous media, should be considered the preferred method. As this study has shown, models created using CFD are capable of accurately simulating the airflow inside large-scale dairy barns; therefore, CFD, once properly validated, can and should be used when solving design problems related to mechanical ventilation systems and conducting research studies seeking to evaluate such systems before they are installed, especially because CFD can be extremely flexible with respect to the modeling and testing of a variety of cases and can save a considerable amount of time and resources. A dairy barn is an extremely dynamic domain in which the microenvironment and macroenvironment are constantly changing. In addition, the cows housed inside dairy barns (not to mention other kinds of livestock kept in similar structures) will typically be moving: capriciously eating, drinking, and constantly lying down or standing up, all of which adds to the complexity. Moreover, dairies generally operate continually, day and night, with farm vehicles transferring feeds, scraping manure, and replacing freestall bedding, all of which can significantly influence the air currents passing through the barn. Nevertheless, in spite of the complexity, the CFD models generated during the course of this study were able to make airflow predictions comparable to those enabled by the physically measured data collected from the barn (and used in the study to validate the CFD analysis). This study also demonstrated the viability of applying porous blocks to represent the cows that would be occupying a typical large-scale computational dairy barn.

**Author Contributions:** Conceptualization, C.Y.C.; Methodology, S.J., H.C. and C.Y.C.; Software, S.J.; Validation, S.J. and H.C.; Formal analysis, S.J.; Investigation, S.J., H.C. and C.Y.C.; Data curation, S.J. and H.C.; Writing—original draft, S.J.; Writing—review & editing, H.C. and C.Y.C.; Visualization, S.J.; Supervision, C.Y.C.; Project administration, C.Y.C.; Funding acquisition, C.Y.C. All authors have read and agreed to the published version of the manuscript.

**Funding:** This research was supported in part by USDA National Institute Food and Agriculture (Joint NSF Cyber-Physical Systems) under Grant No. 67021-34036 and in part by USDA Hatch Grant No. WIS03047.

**Data Availability Statement:** The data presented in this study are available on request from the corresponding author.

**Conflicts of Interest:** The authors declare no conflict of interest.

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
