# Peer review of "Assessment of Airflow Patterns Induced by a Retractable Baffle to Mitigate Heat Stress in a Large-Scale Mechanically Ventilated Barn"

_agriculture, doi:10.3390/agriculture13101910_

Round 1

Reviewer 1 Report

The article is about a CFD simulation study on the effect of baffles on the air flow of a dairy cows large barn.

Overall - even though the scientific problem behind the work is limited to study the effect of retractable baffles on the air flow and the contribution and significance for the community is somewhat limited - the article is well written and organized.

To clarify the problem, Authors should first explain why just one (the central one) of the baffles is retractable and goes to 2 mt instead of the others that goes to 2.2 m form the ground level. Besides, how is operated this retractable baffle, on which schedules/controls?

Some clarifications should be added when it comes to the CFD model, and in particular:

- if the geometry of the study was extended to the whole barn (as it seems from the results on Figure 7;

- which are the boundary conditions (time variable?) considered for the outdoor environment (at least the air temperature at the inlet);

- some goodness-of-fit parameters of the calibration of the CFD model against the measured data should be summarized and presented;

- if the results (from Figure 5 onwards) refer to some steady state conditions and if so, how these were selected;

- some more comments on the Lagrangian particle tracking simulations of Figure 7 should be added in order to understand better why there is this  backflows that occurs closer to the outlet wall, what is the reason of the dissimmetry at the outer wall and if some differences can be appreciated with and without the retractable baffle.

Considering the scope of the work, that is the effect of the baffle(s), once that the wind (? air) velocity is presented on Figure 11, is it possible to derive an indicator of performance on the heat stress (cooling effect on cows, etc.) that is provided by the use of the baffle? This may be really interesting.

Author Response

The article is about a CFD simulation study on the effect of baffles on the air flow of a dairy cows large barn.

Overall - even though the scientific problem behind the work is limited to study the effect of retractable baffles on the air flow and the contribution and significance for the community is somewhat limited - the article is well written and organized.

To clarify the problem, Authors should first explain why just one (the central one) of the baffles is retractable and goes to 2 mt instead of the others that goes to 2.2 m form the ground level. Besides, how is operated this retractable baffle, on which schedules/controls?

>> It was determined by the producer in an effort to increase the wind speed on both sides of the retractable baffle based. Based on our estimates, this height difference does not make any substantial impacts on mitigating heat stress (e.g., the convective heat transfer coefficient).   

Some clarifications should be added when it comes to the CFD model, and in particular:

- if the geometry of the study was extended to the whole barn (as it seems from the results on Figure 7;

>> Yes, that is correct. The readers should be able to understand it based on figures 1 and 2. We added ‘through the entire barn’ in the figure caption to clarify the reviewer’s concern.

- which are the boundary conditions (time variable?) considered for the outdoor environment (at least the air temperature at the inlet);

>> The outdoor environmental conditions do not substantially impact the indoor microclimate conditions, except on extremely windy days. The boundary conditions, as summarized in Table 1, for the CFD simulations were based on experimental measurements. We focused on airflow patterns (not the temperature) in this manuscript. Thermophysical properties of air in the range of measured temperature do not significantly change.

- some goodness-of-fit parameters of the calibration of the CFD model against the measured data should be summarized and presented;

>> Our group published a series of refereed journal manuscripts, which address benchmark problems related to airflow patterns, airflow over blunt bodies including detailed cow geometries, etc. The readers should be able to trace back ‘goodness-of-fit parameters of the calibration against the experimental data’ in the cited references in 3, 7, 9, 10, 12, 13, 14, 15, 24, and 26.

- if the results (from Figure 5 onwards) refer to some steady state conditions and if so, how these were selected;

>> The simulations were based on the steady-state assumption based on the data collected at 1-second intervals over a period 3 minutes at each location to establish the boundary conditions (and all other data points within the barn). The time averaged airflow outcomes hardly changed based on our repeated measurements.

- some more comments on the Lagrangian particle tracking simulations of Figure 7 should be added in order to understand better why there is this  backflows that occurs closer to the outlet wall, what is the reason of the dissimmetry at the outer wall and if some differences can be appreciated with and without the retractable baffle.

>> This particle tracking method is commonly known as discrete phase modeling, and the theoretical backgrounds along with the detailed equations are listed in the literature and software manuals, including ANSYS Fluent Manuals (readily available on websites). The described backflows are quite common due to the significant pressure differences between inside and outside of the fans along the downstream wall (holding the fans), and one can demonstrate such phenomena using the Eulerian method as well. Figure 7 demonstrates a snapshot of these backflows, and the backflow patterns are understandably random and unorganized as they should be, with or without the retractable baffle.

Considering the scope of the work, that is the effect of the baffle(s), once that the wind (? air) velocity is presented on Figure 11, is it possible to derive an indicator of performance on the heat stress (cooling effect on cows, etc.) that is provided by the use of the baffle? This may be really interesting.

>> We agree with the reviewer. It would be interesting to add cows to CFD simulations with heat transfer aspects, which is beyond the scope of the present study. Indeed, we are currently working on this specific topic along with field study with cows (by measuring core body temperature), and the technical contents including the field experimental data (i.e., core body temperature of the cows) are beyond the scope of the present manuscript. 

Reviewer 2 Report

The manuscript addresses a topical issue in the field of agricultural engineering, specifically for the correct design of mechanically ventilated cattle barns. An effective approach to CFD modelling of ventilation solutions with retractable baffles retrieved and extended has been developed, tested and presented.

The results represent useful indications for researchers and technicians, facing the complexity of optimally designing  a so complex kind of buildings.

Some minor revisions have been suggested though, to complete the work and improve its presentation, as the scientific content of the manuscript deserves a very accurate description, to produce the desirable impact on the readers. The specific comments are reported in the file attached.

Author Response

The manuscript addresses a topical issue in the field of agricultural engineering, specifically for the correct design of mechanically ventilated cattle barns. An effective approach to CFD modelling of ventilation solutions with retractable baffles retrieved and extended has been developed, tested and presented.

The results represent useful indications for researchers and technicians, facing the complexity of optimally designing  a so complex kind of buildings.

Some minor revisions have been suggested though, to complete the work and improve its presentation, as the scientific content of the manuscript deserves a very accurate description, to produce the desirable impact on the readers. The specific comments are reported in the file attached.

L33-34 >> Added relevant references in the manuscript as recommended. [3, 7, 22]

L111 >> Corrected “and”

L325, L349, L367 >> Figure captions added: (a) retracted and (b) extended.

Tables 2 & 3 >> Unit added in the cations.

Figure 11 >> Two cases are generally cited in conjunction with the previous figure captions, and they were now defined as CASE (a) and (b). Therefore, Figure 11 (a) though (d) should not confuse the readers.  

 L464 >> We attempted to present the RMSE, in addition to the standard deviation. We could not come up with the meaning outcomes to present the data using RMSE.  

Reviewer 3 Report

This manuscript studies the effect of baffles on airflow patterns in LPCV cattle house based on CFD simulation. The findings are interesting, but there are some unclear points need to explain.

1)The advantage of setting a retractable baffle is not explained. Why set up retractable baffles instead of the five-baffle directly?

2) The article should further study how to regulate the mechanically-ventilation to achieve the effect of energy saving after adding baffles to improve the airflow patterns in AOZ area.

3)Whether there is a field test to verify the conclusion after CFD simulation?

4)The experimental cow house is a regular symmetrical cow house. If only the airflow distribution is studied without involving the thermal environment of the whole cow house, why not consider only half of the study, which can save computing resources and improve computing efficiency?

5) It is necessary to supplement the very important part of CFD model verification, which is carried out in a quantitative way.

6) It is necessary to supplement the simulation parameters of the cow porous media model.

7) The position of rows 5, 6 and 7 (line 290)is not stated in the article "2. Materials and Methods".

8) The data in Table 2 and Table 3 are not used, and are repeated with Figure 10.

Author Response

This manuscript studies the effect of baffles on airflow patterns in LPCV cattle house based on CFD simulation. The findings are interesting, but there are some unclear points need to explain. 

1)The advantage of setting a retractable baffle is not explained. Why set up retractable baffles instead of the five-baffle directly?

>> It is a typical practice by dairy producers and barn designers with limited knowledge of aerodynamics. That is, they first build a barn and observe the air speed pattern and then add the appropriate number of the baffles (four in this case). In this exemplary case, the farm manager recognized that an additional baffle is necessary after operating the barn with the four baffles. This type of retrofitting is common, and this manuscript addresses how computational modeling can minimize time-consuming and labor-intensive trial-and-error and retrofitting efforts. The inexpensive fixed baffles obstruct airflow when fast air speeds are not needed to mitigate heat stress. The retractable baffle can improve airflow at lower ventilation rates and prevent the trapping of stale air between the baffles. Thereby, the most recent addition of the baffle is retractable, recognizing its advantages. We added the advantage of the retractable baffle in the manuscript.

2) The article should further study how to regulate the mechanically-ventilation to achieve the effect of energy saving after adding baffles to improve the airflow patterns in AOZ area.

>> It is well known that each baffle adds about 0.017 inches Hg static pressure, and the number of baffles added to the barn has to be accounted for when choosing the type and number of fans. Each commercial barn is different, and we cannot quantify and generalize the energy usage and the effectiveness of the baffles to mitigate heat stress, which should be presented by cows’ core body temperature (CBT) measurements. This type of field study with cows is beyond the scope of the present computational study.  

3)Whether there is a field test to verify the conclusion after CFD simulation?

>> Yes, we have been developing a controlled environmental study with dairy cows by developing an engineered system by measuring cows’ core body temperature.

4)The experimental cow house is a regular symmetrical cow house. If only the airflow distribution is studied without involving the thermal environment of the whole cow house, why not consider only half of the study, which can save computing resources and improve computing efficiency?

>> The flow patterns are not symmetric as shown in Figure 7. Depending on airspeed, we found that there can be significant backflows and recirculation zones which can significantly the heat removal rate from the cows.

5) It is necessary to supplement the very important part of CFD model verification, which is carried out in a quantitative way. & 6) It is necessary to supplement the simulation parameters of the cow porous media model.

>> Our group published a series of refereed journal manuscripts, which address benchmark problems related to airflow patterns, airflow over blunt bodies including detailed cow geometries, etc. The readers should be able to trace back ‘goodness-of-fit parameters of the calibration against the experimental data’ in the cited references in 3, 7, 9, 10, 12, 13, 14, 15, 24, and 26. We also address the cow porous model in these previous publications.

7) The position of rows 5, 6 and 7 (line 290) is not stated in the article "2. Materials and Methods".

>> In the revised manuscript, we stated that those rows are specified in Figure 3.

8) The data in Table 2 and Table 3 are not used, and are repeated with Figure 10.

>> It may be redundant to present Tables 2 and 3. Therefore, we removed both tables.

Reviewer 4 Report

Minor editing of English language required

Author Response

  1. The main objective of the work is not clearly addressed. Both the title and the aim of the manuscript are too general. The main weakness is that many points and "conclusions" are not new but general knowledge.

>> We addressed the objective of the work clearly in the paragraph starting from L101. The reviewer did not specify the reason why the title, many points, conclusions are not new or general.  

  1. Lack of description of progress in the study of airflow organization forms (mix ventilation、 displacement ventilation 、attachment ventilation、impinging jet)

>> We could not comprehend this statement. Please be specific and constructive.

  1. The grid independence study was described weakly. The information is very limited and questionable for this part.

>> For the length of the manuscript, we did have an appropriate statement in the paragraph starting L238.

  1. No description of software. ANSYS, Star ccm+ ,OpenFOAM?

>> We added the software package in the revised version.

  1. Ar must be considered in non-isothermal flows.

>> We cannot comprehend this statement. What is Ar?

  1. The presentation of results and discussion is unclear and lacks analytical content.

>> Please be specific and constructive.

Round 2

Reviewer 1 Report

The rebuttals were satisfactory.

Author Response

Thank you!

Reviewer 3 Report

The author improved the manuscript and gave response to comments 1 and 8. But I didn't see the response to explain how to consider the other comments. In addition, the "model validation "part is not given to evaluate the accuracy of the model, which is very fatal. 

Author Response

We are not sure why the reviewer did not see the responses to the comments except 1 and 8. We responded to all the comments from 1 to 8. 

For this type of CFD simulations, we often perform a series of experimental measurements and compare the CFD outcomes with the experimental results to validate the model. Otherwise, our finite volume-based method has been verified using a number of benchmark solutions and also other publications with the experimental results over the years. 

Reviewer 4 Report

publish

Author Response

Thank you!